# What Determines the Class of Immunity an Antigen Induces? A Foundational Question Whose Rational Consideration Has Been Undermined by the Information Overload

**DOI:** 10.3390/biology12091253

**Published:** 2023-09-19

**Authors:** Peter Bretscher

**Affiliations:** Department of Biochemistry, Microbiology and Immunology, University of Saskatchewan, Saskatoon, SK S7N 5E5, Canada; peter.bretscher@usask.ca

**Keywords:** immune class regulation, humoral immunity, cell-mediated immunity, antigen dose, efficacy of immunity

## Abstract

**Simple Summary:**

Diverse studies suggest that the increased generation of information, as research intensity increases in a scientific field, leads to an “ossification of the canon” and the neglect of constructive “disruptive” research. I illustrate here how this has occurred in the context of a central question of immunology: what determines the class of immunity induced, cell-mediated immunity or antibodies, when a foreign antigen impinges upon a person or animal? I chose this question for three reasons. It is a basic question and has been the subject of intense research for at least the last three decades. The answer is central to the rational design of strategies of prevention and treatment in diverse areas of medicine related to the immune system: infectious diseases, cancer and allergies. Lastly, the predominant frameworks employed in analyzing this question by immunologists over these last three decades are implausible. Diverse observations on the variables of immunization, which affect the cell-mediated/antibody nature of the ensuing response, are paradoxical within their context. An alternative framework is consistent with these observations. Moreover, this alternative framework brings quantitative considerations to the fore and has considerable implications for treatment and prevention of clinical conditions in infectious diseases and cancer, as I discuss.

**Abstract:**

Activated CD4 T helper cells are required to activate B cells to produce antibody and CD8 T cells to generate cytotoxic T lymphocytes. In the absence of such help, antigens inactivate B cells and CD8 T cells. Thus, the activation or inactivation of CD4 T cells determines whether immune responses are generated, or potentially ablated. Most consider that the activation of CD4 T cells requires an antigen-dependent signal, signal 1, as well as a critical costimulatory signal, initiated when a pattern recognition receptor (PRR) engages with a danger- or pathogen-associated molecular pattern (DAMP or PAMP). Most also envisage that the nature of the DAMP/PAMP signal determines the Th subset predominantly generated and so the class of immunity predominantly induced. I argue that this framework is implausible as it is incompatible with diverse observations of the variables of immunization affecting the class of immunity induced. An alternative framework, the threshold hypothesis, posits that different levels of antigen mediated CD4 T cell interactions lead to the generation of different Th subsets and so different classes of immunity, that it is compatible with these observations. This alternative supports a rational approach to preventing and treating diverse clinical conditions associated with infectious disease and, more speculatively, with cancer.

## 1. Prologue

The focus of this article is on a foundational question of contemporary immunology, namely what determines whether antigens induce predominantly Th1 or Th2 cells under a given set of circumstances? I explain in this prologue why I consider this question more broadly significant in two different ways than might be first apparent from the main body of the text.

There has been considerable interest in recent decades on whether the increased investment in science pays dividends [1,2,3,4]. Most long-time researchers feel that the consequential increased rate of production of information has changed the nature of research. Most would agree with Lewis Carroll: “Now, here, you see, it takes all the running you can do, to keep in the same place. If you want to get somewhere else, you must run at least twice as fast as that!” It is difficult to keep abreast of the new information. Some have persuasively argued that this increased intensity of research has been detrimental; it results in an “ossification of the canon” [2] and a lack of consideration and of appreciation of constructive “disruptive research” [3]. These discussions ring a chord with my experience in immunology and, I gather, with some other immunologists, as well as other researchers in other research-intensive fields. Some judge such “ossification of the canon” to be an inevitable consequence of increased research intensity [2]. I have argued that the style of research fostered by the scientific community is not necessarily static and that changes in the criteria for funding research proposals and in the evaluation of research for publication may ameliorate the situation caused by the information overload [5]. Niels Bohr said: “How wonderful that we have met with a paradox. Now we have some hope of making progress” [6]. The information overload leads to a lack of research focus. I suggest adopting Bohr’s vision, and focusing on finding and resolving paradoxes within the context of the canon, can allow one to transcend the information overload and lead to the evolution of the canon [5]. I think too many researchers are following Lewis Carrol’s reasoning. I suggest that the implication of Bohr’s vision is: “Now, here, you see, it takes all the running you can do, to keep in the same place. If you want to get somewhere else, you may move slower, but in a different direction”. I have adopted, in this article, Bohr’s vision in the context of a particular question in immunology. The information overload tends to result in distinct research silos of researchers in neighboring fields. Observations made in neighboring fields accumulate as unrelated facts. I hope and anticipate that this article will transcend such silos. It is written to be accessible and make sense to individuals with only a minimal knowledge of immunology. I suggest that this article, in this manner, illustrates how to transcend the information overload.

There is a second way in which the ideas and concepts discussed are of broader significance than might appear from the focused context of the main body of the text. A major question in immunology is what circumstances determine the class of immunity generated when an antigen impinges upon the immune system? This is a complex question, as there are two major forms of cell-mediated immunity, namely cytotoxic T lymphocytes (CTL) and distinct T cells, which mediate delayed-type hypersensitivity (DTH), as well as, in people, seven different classes/subclasses of antibody, namely the IgM, IgA, IgE and IgG classes, with there being four IgG subclasses, IgG_1_–IgG_4_. The generation of all these different classes/subclasses of immunity is differentially regulated [7]. Their generation is also associated with the generation of different Th subsets [8]. The question of what different circumstances determine the class of immunity generated can be largely answered if we know the different circumstances determining the generation of each Th subset. These two sets of circumstances are connected if we also know how different Th subsets affect the production of different classes of immunity. However, I focus here on just two Th subsets, Th1 and Th2. This has the advantage of not trying to digest too great a slice of the pie at one time. In addition, more is known about how Th1 and Th2 cells are differentially regulated than for other Th subsets. Furthermore, this limitation of our considerations of Th1 and Th2 cells is not so great a limitation as might at first appear, for several reasons. I discuss the nature of the *decision criterion* controlling the Th1/Th2 phenotype of the response in terms of the frameworks most prominent in the literature over the last twenty/thirty years [9,10,11,12,13], as well as in terms of the alternative I favor [14,15]. I suggest and will argue that these prominent frameworks are implausible as there are so many paradoxes within their context [7,16]. Moreover, most of the analysis of what circumstances favor the generation of Th cells belonging to subsets other than Th1 and Th2 cells employ these same prominent frameworks that I suggest are implausible. I will indicate the basis of my skepticism. Thus, the analysis of what is the nature of the decision criterion controlling the Th1/Th2 phenotype of a response has broader significance than just this criterion.

## 2. Frameworks for Envisaging What Controls the Th1/Th2 Phenotype of a Response

As indicated above, the class of immunity generated upon primary immunization depends upon the Th subset generated when antigens activate naïve, mature CD4 T cells [8]. All models for the nature of the decision criterion controlling the Th1/Th2 phenotype of primary responses are cast within the context of a model for what is required to activate naive CD4 T cells. Indeed, the predominant model for the nature of this decision criterion is based upon the model predominantly held as to what is required to activate naïve CD4 T cells. This is the DAMP/PAMP model [17,18,19,20]. The activation of a naïve, mature CD4 T cell is envisaged to require an antigen-dependent signal, signal 1, and a critical costimulatory signal, mediated by costimulatory (CoS) molecules on an antigen-presenting cell (APC) interacting with their counter receptors on the CD4 T cell. The expression of the critical CoS molecules on the APC is envisaged to be upregulated when a pattern recognition receptor (PRR) of the APC interacts with a danger- or pathogen-associated-molecular pattern (DAMP/PAMP). Thus, the activation of CD4 T cells, and so virtually all immune responses, is envisaged to require a DAMP or PAMP signal. Moreover, there are diverse pairs of CoS molecules and their counter receptors whose interaction can constitute a CoS signal [21]. In general, CoS signals, delivered to a responding T cell, modulate the effects of signal 1 in terms of the differentiation of the responding CD4 T cells that conjointly receive signal 1. 

## 3. The Grounds for the PAMP Model for the Activation of CD4 T Cells

I have to distil the arguments made by the proponents of these models to consider their plausibility and to provide context. These models were initiated by Janeway in 1989 [17]. He noted that T cells, specific for self-antigens sufficiently present in the thymus, are eliminated by a mechanism [22] first proposed by Lederberg [23], resulting in what is referred to as central tolerance. Those T cells uniquely specific for foreign antigens emigrate from the thymus into the periphery; in addition, others emigrate that are specific for self-antigens insufficiently present in the thymus to cause their elimination. Such antigens are often more prevalent in the periphery and are then referred to as peripheral self-antigens. Insulin is a prototypical peripheral self-antigen [24]. As I have recently discussed elsewhere, Janeway’s proposal has led to the contemporary beliefs that the activation of a CD4 T cell requires a critical CoS signal and, in the absence of this costimulatory signal, antigen can inactivate the CD4 T cell [25]. This is a really radical and, to me, an implausible proposal, as I shall later explain. As the large majority of immune responses require the activation of CD4 T cells, Janeway’s proposal means that all these immune responses require the presence of a PAMP; whether CD4 T cells are activated and so an immune response is initiated, or are inactivated by an antigen, does not depend upon whether they are specific for foreign or self-antigens but whether or not they encounter the antigen in the context of a PAMP. 

## 4. The Grounds for the DAMP Model for the Activation of CD4 T Cells

Matzinger, five years after Janeway’s proposal, pointed out that grafts between mice belonging to different strains are rejected but, being of vertebrate origin, are not expected to bear PAMPs [19]. Rejection of such grafts is not readily explicable in Janeway’s view. Matzinger proposed that such grafting causes stress and the generation of a danger signal, following the upregulation of, for example, stress proteins. The danger signal is envisaged to upregulate critical CoS molecules on APCs and to be required for the activation of CD4 T cells. Matzinger also considered other grounds in support of her danger model [19,20]. I have recently discussed elsewhere why I do not find these grounds compelling [25]. 

## 5. The Plausibility/Implausibility of the DAMP/PAMP Models

In my view, Janeway’s PAMP and Matzinger’s DAMP models are very similar. They posit, respectively, that a PAMP signal and a DAMP signal are required to activate CD4 T cells and so for the generation of all immune responses. This contrasts with the classical view that the immune system responds usually only against foreign but not self-antigens, in accord with its attribute of self–nonself discrimination. I argue that there are many foreign, vertebrate antigens that, when injected with a sharp needle, are immunogenic [7]. How is the envisaged DAMP/PAMP signal generated in such cases? Why are immune responses generated when goats are administered red blood cells from another goat whose red cells are chemically different from those of the immunized goat [26]? Such red cells are not expected to bear PAMPs. How do mice respond so readily to various foreign vertebrate red blood cells, such as those of sheep? The suggestion that injection with sharp needles causes a sufficient danger signal is not credible. People, particularly children, often fall and graze themselves by accident, but such incidences are not generally associated with autoimmunity. The evidence in my judgement is overwhelmingly consistent with the view that the immune system expresses the attribute of self–nonself discrimination, as has been classically supposed by immunologists for over a century [23,27,28], at both the levels of central [22] and peripheral tolerance [29]. The prominence and popularity of the DAMP/PAMP view surprises me. This, of course, does not mean that DAMP/PAMP signals are not physiologically significant. It makes evolutionary sense that the detection of DAMPs and PAMPs increases the speed and intensity of immune responses. Thus, inflammatory signals can increase the size and tempo of immune responses. I am arguing that PAMPS and DAMPs do not play the pivotal roles envisaged by Janeway and Matzinger [9,10,11,12,13].

## 6. The Quorum Model for Lymphocyte Activation, Including the Activation of CD4 T Cells

Cohn and I proposed in 1970 a model for how antigens interact differently with mature lymphocytes to result in their activation or in their inactivation [29]. The proposal was that the activation of all lymphocytes requires antigen-mediated lymphocyte cooperation, whereas the interaction of antigens with a single lymphocyte would inactivate the lymphocyte. The model was proposed for two reasons. It resolved various paradoxes then apparent, as discussed elsewhere [30]. Secondly, it provided an explanation of peripheral tolerance consistent with the idea that tolerance to a peripheral self-antigen follows its first presence early in development, or in the “history” of the individual, at a time before lymphocytes have begun to be generated, and the continuous presence of the antigen thereafter. This proposal, made by Burnet and Fenner in its initial form [27] and later modified by Lederberg [23], is referred to as the historical postulate, see Figure 1. According to our model, the first lymphocyte specific for a peripheral antigen, such as insulin, will be a loner and so inactivated; further lymphocytes specific for insulin will be inactivated as generated, one or a few at a time. Lymphocytes specific for a foreign antigen will accumulate in the absence of the antigen. When the foreign antigen impinges upon the individual, it can mediate the lymphocyte interactions leading to immune responses, see Figure 1. We proposed that the interaction of antigen with the lymphocyte’s antigen-specific receptors generates signal 1, which results in the inactivation of the lymphocyte when generated alone. The responding lymphocyte needs a second signal, signal 2, to be activated. The generation of signal 2 requires the “helper” lymphocyte to also recognize the antigen, triggering the generation and delivery of signal 2 to the responding lymphocyte. We suggested that signal 2 was mediated by short-range molecules, which are now referred to as cytokines, and/or interactions between the surface membrane of the interacting cells, now referred to as CoS signals [29,30]. One reviewer suggested it would be helpful to know how signal 2 is mediated in detail at the molecular level. I think signal 2 is surely different at the molecular level for different types of lymphocytes. In addition, it is not necessary to know such detail in testing The Two Signal Hypothesis, a proposal primarily at the level of the system. Moreover, a mistaken or only partial identification of signal 2 in molecular terms may lead to observations that are erroneously interpreted as evidence against the hypothesis. It should be added for clarity that, in more recent times, the CoS signal is often referred to as signal 2 and the delivery of cytokines as signal 3. 

This model was originally referred to as the Two Signal Model for lymphocyte activation. The findings that the activation of B cells and of CD8 T cells requires activated CD4 T cells, and that the antigen inactivates the B and CD8 T cells in the absence of such help, was in accord with the model. The original model [29], and its more modern formulations [30,31], predicted that the activation of CD4 T lymphocytes also requires lymphocyte cooperation and, in the absence of such cooperation, the antigen inactivates the CD4 T cell. Thus, an essential feature of the model is that the activation of all lymphocytes requires antigen-mediated lymphocyte cooperation. Other models for the activation of CD4 T cells were later proposed, involving two signals, such as those put forward by Janeway [17,18] and Matzinger [19,20]. However, as already indicated, the biological significance of these and our models are profoundly different. The former models incorporate the idea that the immune system responds only when the PAMP of an infectious agent is present or when a danger signal is generated. Our model embraced the idea that a healthy immune system responds to foreign but not peripheral self-antigens, and so it displays the attribute of self–nonself discrimination [29]. The description of all these models as two-signal models of CD4 T cell activation, despite their biologically significant differences, led to some confusion. I now use a different terminology to avoid this ambiguity. Our model posits that a minimum number of lymphocytes, a “quorum” of lymphocytes, is required to activate lymphocytes, including CD4 T cells, under optimal conditions of, for example, antigen concentration. I now refer to our idea as the Quorum Model of Lymphocyte Activation [31].

I have recently reviewed the evidence for this model, with emphasis on the activation/inactivation of CD4 T cells. I therefore only summarize certain important conclusions here to provide context [25].

The activation of both B cells [32,33] to produce antibodies and of CD8 T cells [34] to generate CTL requires activated T helper cells; in the absence of such help, antigen inactivates the B cells [35,36] and the CD8 T cells [37]. These observations are in accord with The Quorum Model and lead to an appreciation of the pivotal role of CD4 T cells in the activation of other classes of lymphocytes. I argue that the evidence supports the proposal that the activation of CD4 T cells requires CD4 T cell cooperation, with a B cell acting as an APC, mediating this cooperation [25,30,31]. However, I should stress, for clarity, that this is a minority view.

The elucidation of circumstances leading to autoimmunity supports The Quorum Model. Weigle, in the 1960s, conducted a series of experiments that appeared to define conditions under which autoimmunity might be generated [38,39]. He found that in certain circumstances, he could induce autoantibodies by immunizing with a foreign antigen that cross-reacted with a self-antigen. Immunization of rabbits with turkey thyroglobulin resulted in the production of antibodies that reacted with both turkey and rabbit thyroglobulin. This type of observation finds a natural explanation in terms of The Quorum Model [31], once it is appreciated that rabbit thyroglobulin must be a peripheral antigen and so lymphocytes specific for it are exported from primary lymphoid organs. This observation, in terms of modern knowledge, means there are B cells with receptors specific for both rabbit and turkey thyroglobulin in mature rabbits. Rabbit thyroglobulin is unable to activate these B cells due, presumably, to a lack of a quorum of lymphocytes to activate CD4 T helper cells, whereas there is a quorum for turkey thyroglobulin, this antigen being more foreign. The pertinence of Weigle’s findings to the generation of autoimmunity was confirmed when it was found that infection with group A streptococci could induce rheumatic heart disease, associated with antibodies that bound to both a component of the bacteria and of heart tissue [40]. Furthermore, such an infection leads to the activation of CD4 T cells reactive to both heart tissue and the bacteria [41]. As outlined elsewhere, this is understandable if CD4 T cell activation requires a quorum of lymphocytes [31]. It is much less readily explicable on the DAMP/PAMP model of CD4 T cell activation. In particular, the Quorum Hypothesis explains why group A streptococci and heart tissue must cross-react at the level of CD4 T cells for CD4 T cell autoimmunity to be induced, whereas such cross-reactivity is not anticipated to be necessary on the DAMP/PAMP Model. The incidence of autoimmunity would be expected to be much more prevalent on the latter model. 

One study, carried out by Janeway and colleagues, is highly paradoxical for the PAMP Model and provides evidence strongly supporting the Quorum Model in the context of the activation of CD4 T cells. These researchers found conditions under which CD4 T cells specific for *mouse* cytochrome C (MCC) could be raised in *mice* [42]. Immunization with MCC in complete Freund’s adjuvant (CFA) was ineffective. The same immunization protocol was effective when the immunized mice in addition passively received MCC-specific, activated B cells. Janeway and colleagues interpreted their observations as showing that the activation of CD4 T cells could be facilitated by B cells that had been activated by antigens in a Th-cell-dependent fashion [42]. I agree. They did not, as far as I can tell, address why their observations are not paradoxical within the context of Janeway’s PAMP model. 

## 7. The General Importance of Immune Class Regulation

Many observations in people and in animal models of human disease have led to an appreciation of the critical role of the class of immunity, generated against the pertinent antigens, to clinical outcome. The first example was leprosy. It was recognized that there is a spectrum of disease; in tuberculoid leprosy, a predominant and stable cell-mediated response is manifest, and pathogen load and pathology are minimal, whereas in lepromatous leprosy and “borderline” disease, in which a predominant IgG antibody and a mixed cell-mediated/IgG antibody response respectively occurs, severe pathology co-exists with a high burden of the bacteria [43]. Many observations appear to show that protection against the pathogen causing tuberculosis requires a Th1 response [44]. It has been argued that some observations are paradoxical in the context of “this central dogma” [44]. We have argued elsewhere that such observations can be accommodated by a hypothesis that only a sufficiently strong and predominant Th1 response against the pathogen is protective [45]. Another topical and pertinent pathogen is HIV-1. A small number of individuals infected with HIV-1 generate a sustained Th1, CTL response and do not seroconvert or suffer progressive disease [46]. The existence of these “elite controllers” leads some researchers, including myself, to argue that effective vaccination must guarantee a stable cell-mediated response upon infection. We therefore explored how this might be achieved. The most prominent animal model of an infectious disease, caused by an intracellular pathogen uniquely susceptible to cell-mediated attacks, is the mouse model of cutaneous leishmaniasis [47,48]. The outcome of infection of a mouse with a million *Leishmania major* parasites depends on the strain to which the mouse belongs. Resistant mice generate a sustained Th1 response, whereas the response of susceptible mice develops in time a predominant Th2 response associated with production of the IgG antibody and loss of parasite control [43]. However, several different maneuvers result in a sustained Th1 response in susceptible mice upon challenge with a million parasites; they also result in resistance [49,50,51]. Such findings strongly support the idea that the association of a Th1 response with resistance is not merely correlative but causative. One means of guaranteeing a sustained Th1 response in “susceptible mice” is by infecting them with a low number: three hundred parasites. This low number not only generates a sustained Th1 response but, in time, what we call a Th1 imprint: a sustained Th1 response, and so resistance to a subsequent and normally lethal challenge of a million parasites. We refer to this as the low-dose vaccination strategy [51]. We also showed that this strategy works with mycobacteria [52,53], the phylum that contains the intracellular pathogens that cause leprosy and tuberculosis. Buddle and colleagues showed that infection of cattle, with about a million-fold lower number of BCG than the number usually employed, provides dramatic protection against tuberculosis in an experimental model of the disease [54]. This protection contrasts with the lack of reliable protection by immunization with the standard dose in humans [55] or in cattle [54]. 

While the uniquely pivotal role of the class of immunity is clear in protection against some intracellular pathogens, there is less of a consensus, and less compelling evidence, for the unique role of cell-mediated immunity in providing protection against cancers. Nevertheless, it is worthwhile to carefully assess the evidence as it now exists. It is helpful if I explicitly state at the beginning the conclusions I think are plausible. Most researchers would agree that a sufficiently strong cell-mediated response is protective against cancer [56], whereas *naturally produced* antibody is rarely, if ever, protective. Progressive cancer will occur if the cell-mediated response is too weak, either because the cancer-associated antigens are insufficiently immunogenic to generate a strong cell-mediated response, or because they generate an antibody response, associated with the downregulation of the cell-mediated response [57]. I first consider animal models of human cancer. These are of interest but should not be considered as necessarily providing reliable models for human cancer. The relevance of conclusions drawn from animal studies, most of which employ transplantable tumors, to spontaneously occurring human cancer, is not obvious [57]. However, I think that various studies in animal systems are suggestive. To make this case, I first summarize various animal studies before considering their potential pertinence to human cancer. 

George Klein, one of the founders of modern tumor immunology, wrote, in 1968, in the context of efficacious vaccination against cancer: “It will be most important to establish what variables of the immunization……..dosage, route of administration, and timing, are critical to achieve the objective, which seems to be a stimulation of host cell-mediated rejection with minimum risk of antibody-mediated enhancement” [58]. This statement clearly reflects the view that cell-mediated immunity can be effective against cancers, whereas the production of IgG antibodies is often associated with tumor progression [59]. The transplantable tumors employed were either generated by infection with oncogenic viruses or occurred in animals exposed to carcinogens. Systematic studies [60,61] by Robert North, decades later, were designed to understand the basis of the phenomenon of concomitant immunity that is observed in tumor immunology: injection of a lethal dose of tumor resulted in resistance to a second, normally lethal challenge of the tumor, injected a few days after the primary challenge, at a distal site [62]. The primary tumor, in contrast, grew progressively. Robert North clearly expressed his belief and hope that an understanding of the basis of concomitant immunity and its decay would lead to effective immunotherapy against cancer. His studies mapped out the transient appearance of protective T cells shortly after the injection of a lethal dose of tumor and the appearance of tumor-specific “suppressor T cells” as the presence of protective cells declined. North encapsulated his solution to the enigma of the existence of concomitant immunity and of tumor progression, following a lethal injection of the tumor, as “too little (protective immunity) too late”. Most now regard North’s work as demonstrating the importance of “Treg cells” in undermining the generation of protective immunity [63,64,65,66]. We shall return to the subject later, with a different proposal as to the significance of North’s findings. 

## 8. Variables of Immunization Affecting the Th1/Th2 Phenotype of the Ensuing Response: Salvin’s Findings

Significant studies on these variables started in the 1950s. I wish to summarize a nexus of classical, related observations. I argue that these represent valid generalizations about how immune responses are regulated. However, these generalizations are not or are insufficiently recognized today. I suggest this is in part because they are paradoxical in terms of today’s most commonly held frameworks. I will later justify this assessment.

Salvin examined in the 1950s how the dose of antigen and time after immunization affected the cell-mediated/IgG nature of the ensuing response, see Figure 2 [67]. Low doses of antigen lead to an exclusive cell-mediated, DTH response. Medium doses resulted in a more rapid DTH response that declines as the IgG antibody is produced. A high dose gives rise to even more rapid responses, and the exclusive, DTH phase is shortened or even eclipsed. I argue below these observations are related to several others that also appear to be of a general nature and to be critical in understanding the broad features of how immune responses are regulated.

### 8.1. The Generality of Salvin’s Findings

Salvin’s findings have been found to be generally true in diverse circumstances when an antigen is administered parenterally, i.e., by a route that involves breaking the skin. Mucosal immunization, involving an antigen entering the body without skin rupture, such as via the gastrointestinal and urogenital tracts, and through inhalation, often result in different classes of immunity, and different Th subsets [68] and will not be discussed here, for reasons outlined in the Prologue. Salvin’s observations appear to be generally valid when an antigen is administered by different parenteral routes, such as the intravenous [69] and subcutaneous routes [70], in different species of host (e.g., in mice [69] and cattle [54]) for antigens administered without any adjuvant [71] or with different adjuvants [69]. Interestingly, Salvin employed purified proteins as his antigens [67]. These are not characteristic of most natural antigens. Natural antigens can be more complex in one and/or two ways than the “simple antigens” of Salvin. “Complex antigens” may be chemically more complex, and they may replicate. Examples of a class of more complex, non-replicating antigens, much studied by immunologists but not so obviously reflecting natural antigens, are foreign, vertebrate erythrocytes, such as sheep red blood cells (SRBCs) administered to mice [69,70]. Consider such a complex antigen in which the different antigen components exist in different amounts. One would expect these antigens, if isolated from the complex antigen and administered separately in amounts corresponding to their prevalence in the complex antigen, to generate responses of different Th1/Th2 phenotypes. If this were the case, it is hard to understand how the cell-mediated/IgG antibody nature of the “collective response” to the different antigen components of SRBCs would follow generalizations based on Salvin’s observations. However, the cell-mediated/IgG antibody nature of this collective response to different doses of SRBCs in mice does follow the tendencies shown in Figure 2 [69,70]. Why is this?

I suggest, before considering this, that it is best to address other observations that I think bear on this question. Firstly, we have examined how the Th1/Th2 phenotype of responses to two non-cross-reacting antigens, administered together in the same syringe, affect each other’s Th1/Th2 phenotype. They do not. For example, we chose two antigens, Q and R, and a dose of Q that, when given alone, generates a predominant Th1 response and a dose of R that, when given alone, generates a response with a substantial Th2 component. The Th1/Th2 nature of the immune responses in mice simultaneously immunized with both antigens is identical to the response in mice similarly immunized with just one antigen [69]. We refer to this finding as reflecting the *Principle of Independence*: the Th1/Th2 phenotype of responses to non-cross-reacting, even when occurring in the same lymphoid organ, are determined independently. (Exceptions to this rule, that occur, for example, in animals experiencing overwhelming parasite infections, reflect these exceptional circumstances [71].) An implication of these observations, supporting the principle, is that the Th1/Th2 phenotype of the response to one component of a complex antigen seems to be regulated to be similar to the Th1/Th2 phenotype of the response to other components. We refer to this phenomenon as *coherence* [72]. Coherence is, in a related context, an acknowledged phenomenon. We know the class/subclass of antibodies produced to different epitopes of an antigen tends to be coherently regulated. Thus, the antibody produced at any one time against an antigen tends to belong to the same class/subclass; when this class/subclass changes as the immune response evolves with time, it tends to do so coherently. How this coherence is achieved is now evident. B cells specific for all the diverse epitopes of an antigen all present the same diverse peptides of the antigen upon endocytosing and processing the antigen. Thus, these diverse B cells receive signals from the same population of T helper cells [72]. It is of course the nature of the Th phenotype of this T cell population that determines the class/subclass of the antibody produced by the descendants of the activated B cell. Thus, the known mechanism of B cell activation accounts for coherence at the level of the class/subclass of antibody produced [72]. We shall later consider how coherence at the level of Th subsets may be achieved. The mechanism proposed at the Th level is closely related to this mechanism, accounting for coherence at the level of B cells.

These considerations on coherence allow us to begin to understand how complex, non-replicating antigens obey the “Salvin laws”. I discuss other classical observations below that are helpful in understanding that many, but not all, replicating antigens also obey the “Salvin laws”. 

### 8.2. Immune Deviation and Th Subset Imprinting 

As already acknowledged, different classes/subclasses of immunity tend to be exclusively generated. This tendency is seen in Figure 2. Low doses of antigen only generate a cell-mediated response. Medium doses initially generate an exclusive cell-mediated response, that declines as IgG antibody is produced. What might be the basis of this exclusivity? Studies in the 1960s and 1970s provided suggestive observations.

Asherson and Stone demonstrated in the mid-1960s that mice, induced to produce IgG antibodies to an antigen, could no longer be induced to express DTH to the antigen upon a challenge that generated such a response in naïve mice [73]. We refer to this state as a state of *humoral immune deviation*. Parish [74,75], exploiting a system developed by Mitchison [76], found conditions leading to a state in which the immune response to the antigen was locked into a cell-mediated mode. Both Mitchison and Parish repeatedly administered, several times a week, for several weeks, antigen to mature, i.e., immunologically competent, rodents, each rodent receiving the same dose of antigen each time but different rodents receiving different doses. They then challenged all the rodents with an antigen challenge that in naïve, mature rodents generated an IgG response. Control rodents, injected with saline in the pre-challenge schedule, produced IgG antibody. Those repetitively given a low dose of the antigen in the pre-challenge schedule produced less IgG antibodies and those pre-exposed to medium doses made an increased IgG response [74,75,76]. Parish also measured the state of DTH to the antigen at the time of the challenge. He found no DTH in rodents pre-exposed to saline and minimal expression in those rodents pre-exposed to medium doses of antigen that produced an IgG memory response on the challenge. However, those rodents pre-exposed to low doses of antigen, and that produced substantially less IgG antibodies on the challenge, expressed DTH to the antigen [74,75]. It appeared that the immune response to this antigen was locked into a cell-mediated mode. These studies, of the 1960s, were carried out with “simple antigens”.

What physiological situation might correspond to the Mitchison/Parish protocol involving the *repeated* administration of non-replicating antigens to immunocompetent animals? Consider infection by a rapidly growing parasite. Such infection is known to invariably result in the production of IgG antibodies independently of the infecting dose. Even infection by a low number of rapidly growing organisms rapidly results in a high antigen load. Thus, the production of the IgG antibody can, in this manner, be readily reconciled with “Salvin’s laws”, depicted in Figure 2. Consider infection by a slowly growing pathogen or other entity. Infection with a low number of such entities might well produce a pattern of antigen stimulation similar to that provided by a repetitive low dose of a simple antigen, given according to the Mitchison/Parish protocol, and result in a stable cell-mediated response. This possibility led us to consider two different potential consequences. Infection with low and high numbers of slowly growing organisms/entities will result in responses in accordance with “Salvin’s laws”. We have successfully tested this proposition in mice with transplantable tumor cells [57], with mycobacteria [52,53] and with the protozoan, intracellular parasite, *Leishmania major* [51,77]. In all cases we found that “Salvin’s laws” held. Our studies with *L major* were particularly extensive and interesting, and so we elaborate upon them below, in the next section.

The second potential consequence followed from Parish’s studies. A chronic cell-mediated response appears to lock the immune response into a cell-mediated mode. We illustrate our various studies with those we carried out with *L major* in *susceptible* BALB/c mice. 

As explained above, infection of BALB/c mice with a million *L major* parasites rapidly results in a predominant Th2 anti-parasite response and loss of parasite control. We demonstrated that infection of BALB/c mice with the low number of 300 parasites results in a stable Th1 response. We challenged these mice two months post-infection with a million parasites; they generated a stable Th1 response associated with parasite control. Thus, we could render “susceptible mice” resistant [51]. We refer to this as the low-dose vaccination strategy of Th1 imprinting. We have successfully tested this strategy with transplantable tumors in mice [57] and with mycobacteria in very young [53] and mature mice [52]. This strategy thus seems to me to be broadly applicable. However, one recognized barrier to designing a universally efficacious vaccination protocol is the genetic diversity of human and animal populations. Moreover, genetic diversity may also legitimately raise questions as to the university of “Salvin’s laws”.

## 9. “Salvin’s Laws” Apply in Diverse Strains of Mice in Their Response to *L Major*

We injected mice of diverse strains in an identical fashion with different numbers of *L major* parasites and characterized the Th1/Th2 phenotype of the ensuing responses. We found that the nature of the responses of all strains of mice could be described by a rule. Infection with a sufficiently low number of parasites resulted in a stable Th1 response, and infection with a higher number resulted in a response that, in time, developed a substantial Th2 component. We could define for any mouse strain a *transition number*, *N_t_*. Infection with a number of parasites below N_t_ resulted in a stable Th1 response, and with a number above N_t_, in a response that in time developed a substantial Th2 component. Infection with a number considerably above N_t_ resulted in a response that rapidly developed a predominant Th2 phenotype. The transition number varied over a 100,000-fold range, being about 500 for BALB/c mice and about 5 × 10^7^ for CBA mice [77]. These findings are interesting from three perspectives. Firstly, they demonstrate that “Salvin’s laws” generally hold. Secondly, genetic differences appear to be responsible for the wide diversity in N_t_. This led us to reflect on the fact that only about 1% of HIV-1-infected people, the elite controllers, generate a stable Th1, CTL response and so resist the infection [46]. It seems that our genetic diversity ensures diversity in the nature of our responses against newly arising pathogens, and thus contributes to our survival as a species. Lastly, the generality of “Salvin’s laws” allows one to envisage strategies of universally efficacious vaccination. Although a pathogen would not, for ethical reasons, be employed for vaccination, we can illustrate a strategy in the context of the mouse model of cutaneous leishmaniasis. Suppose we infect genetically diverse mice with a number of *viable* leishmania parasites below the transition number of any individual infected. All mice are expected in this case to develop a chronic Th1 response and in time a Th1 imprint, and so become resistant to what would otherwise be a normally pathogenic infection [77,78].

## 10. Models and Ideas on What Controls the Th1/Th2 Phenotype of an Immune Response

### The DAMP/PAMP-Centric View

Ideas on what determines the Th subset predominantly generated are naturally usually cast in terms of models for the activation of CD4 T cells. A number of articles in the most prominent journals make the case that the nature of the DAMP/PAMP signals are critical, as well as local factors, often reflecting the route of antigen impingement, particularly when mucosal [9,10,11,12,13]. I merely sketch the kind of observations that have been influential, focusing on in vitro studies and in vivo studies involving parenteral routes of immunization/infection.

An early and influential finding was that the presence of dead *Listeria monocytogenes* could stimulate the in vitro production by macrophage-like cells of the cytokine, IL-12, that in turn facilitated the generation of Th1 cells [79]. The stimulation of the production of IL-12 presumably depended on a PRR/PAMP interaction. Many studies, in admittedly rather artificial in vitro systems, showed that other cytokines could preferentially facilitate the generation of Th1 and Th2 cells. Thus, the presence of IL-4 facilitated the generation of Th2 cells [80] and IFN-γ facilitated the generation of Th1 cells [81]. The significance of the physiological role of cytokines was strongly supported by in vivo studies carried out in the mouse model of cutaneous leishmaniasis. Thus, the administration of an antibody that neutralizes the activity of IL-4, around the time of infection of susceptible mice with a million parasites, abrogated the Th2 response and resulted in a sustained Th1 and control of parasite load [82]. The administration of antibody that neutralizes the activity of IFN-γ around the time of infection of resistant mice with a million parasites, abrogated the Th1 response and resulted in a sustained Th2 and loss of control of parasite load [83]. An important role of cytokines in determining the Th subset generated is evident from these and other observations. A question arising from this conclusion is what circumstances are responsible for the production of these different cytokines and how is such production regulated? This is a question we will later return to in a rather general way. In addition, PAMP and DAMP signals, besides having a potential role in stimulating cytokine production, affect the expression of costimulatory molecules [9,10,11,12,13], and in this way could be important in influencing the Th subset generated. We shall discuss shortly why I think it is implausible that the Th1/Th2 phenotype of immune responses is generally determined by such means.

## 11. The Threshold Hypothesis and Its Plausibility

This hypothesis, the one I favor, is based on The Quorum Model for the activation of CD4 T cells [31]. The Threshold Hypothesis posits that, when the antigen-mediated interaction between CD4 T cells, required for the activation of the responding CD4 T cell, is weak, the responding CD4 T cell gives rise to Th1 cells; when robust, the responding CD4 T cell gives rise to Th2 cells. In other words, a lower *threshold* of CD4 T cell interaction is required to give rise to Th1 than Th2 cells [7,14]. A major reason for this quantitative proposal, formulated almost 50 years ago, was the fact that it could account for the variables of immunization then characterized as affecting the Th1/Th2 phenotype of the response, as outlined below. The same reasons hold today, in addition to the successful testing of many predictions of the Threshold Hypothesis, as briefly described below. 

## 12. How the Threshold Hypothesis Explains the Variables of Immunization Affecting the Th1/Th2 Phenotype of a Response

Pearson and Raffel pointed out in the 1960s that certain antigens were able to induce cell-mediated immunity but not readily detectable antibody. These antigens were either small in size or larger, but being only a slight modification of self. These authors proposed that such antigens are minimally foreign, and it is this characteristic that allows them to be immunogenic only for a cell-mediated response [84]. There will be fewer CD4 T cells specific for such antigens than for more foreign antigens. Even in the presence of amounts of antigen optimal for mediating CD4 T cell interactions, only weak CD4 T cell interactions will take place. Thus, according to the proposed threshold mechanism, Th1 cells will be predominantly generated. I suggested such antigens are susceptible only to cell-mediated attacks, providing a physiological reason underlying this mechanism [14]. This possibility may also explain why cancers are preferentially susceptible to cell-mediated attacks [14,57].More foreign antigens, for which there are naturally more CD4 T helper cells, can induce the generation of Th1 or Th2 cells, depending upon the circumstances of immunization. Immunization with low doses of antigen, well below that optimal for mediating CD4 T cell cooperation, will initially only support weak CD4 T cell collaboration and so the generation of Th1 cells, see Figure 2. It is known that foreign antigens cause their corresponding CD4 T cells to multiply. Thus, so long as the level of antigen is sufficiently sustained, the interaction between the CD4 T cells will become stronger with time, thus explaining why the response evolves with time from an exclusive Th1 towards a Th2 phenotype, see Figure 2. Immunizing with higher levels of antigen, more optimal for mediating CD4 T cell collaboration, results in even more rapid responses [7,14].

## 13. Support for the Threshold Hypothesis and Paradoxes within the Context of DAMP/PAMP-Centric View

The strongest and most specific prediction of the threshold mechanism is that, in a situation where an antigen challenge results in a primary, predominant Th2 response, an appropriate and partial depletion of CD4 T cells will result in a modulation of the immune response towards a Th1 mode. We have tested this prediction in diverse systems in responses against foreign, vertebrate and therefore PAMP-free antigens. These systems include intact mice, lethally irradiated mice reconstituted with different populations of unprimed spleen cells to determine what different cell populations support the generation of Th1 and of Th2 cells, and in vitro systems, some employing polyclonal, antigen-specific CD4 T cells and others employing TcR transgenic CD4 T cells, as reviewed elsewhere [15]. I consider the successful testing of this prediction in so many different systems to be compelling. In addition, we showed that the number of CD4 T cells and level of antigen interdependently affected the Th1/Th2 phenotype of the response in the anticipated manner [85,86]. Furthermore, others have shown that the partial depletion of CD4 T cells in BALB/c, close to the time of infection with a million *L major* parasites, modulates the long-term response from a Th2 to a Th1 mode, allowing the mice to contain the pathogen [49,50], again confirming the prediction.

These observations, besides supporting the threshold mechanism, are very difficult to reconcile with the DAMP/PAMP-centric view. Firstly, in all these systems, the nature of the antigen challenge was the same in the different experimental groups, yet the Th1/Th2 phenotype of the response changed. We employed in our studies foreign, vertebrate, PAMP-free antigens, whereas the studies with *L major* involved protozoan parasites anticipated to bear PAMPs. The parallel nature of findings for PAMP-bearing and PAMP-free antigens suggests a common mechanism not involving PAMPs and not involving DAMP signals associated with the mode of immunization [16]. 

The similar antigen dose dependence of the Th1/Th2 phenotype of the response against foreign, vertebrate, PAMP-free antigens, including transplantable tumors [57], and for antigenic entities carrying different spectra of PAMPs, such as mycobacteria [52,53,54] and *L major*, a protozoan parasite [51], is again best explained by a common mechanism not involving PAMPs and not involving a DAMP signal playing a pivotal role. The similar and frequent development of the immune response to foreign, vertebrate, PAMP-free antigens, to mycobacteria and protozoa, as already described, and to HIV-1 [86] from am exclusive cell-mediated mode to one containing a substantial humoral, Th2 component is explicable on the Threshold Hypothesis. However, it is paradoxical in the context of PAMP/DAMP framework as the PAMPs do not change with time after infection [16]. I find it difficult to understand how these considerations are not found to be worthy of serious consideration by the immunological community. I have come to think it may be in part because many observations, not discussed above in any detail, appear to fit in with the PAMP/DAMP-centric view so naturally. There are so many reports on how the cytokine environment, in which naïve CD4 T cells are activated, affects the Th subset to which the activated Th cells belong [9,10,11,12,13]. All textbooks contain figures or tables indicating which cytokines facilitate the generation of activated Th cells belonging to different subsets. Most envisage that PAMP/DAMP signals influence this cytokine environment. I refer to these ideas as The Cytokine Milieu Hypothesis. The evidence supporting the threshold mechanism is perhaps not creditable given the diverse reports supporting The Cytokine Milieu Hypothesis. Is reconciliation possible? 

## 14. The Role of Cytokines in Controlling the Th1/Th2 Phenotype of the Response

The studies supporting the role of cytokines, described above, are *often* obtained in rather artificial, experimental systems. In many studies, the effects of a recombinant cytokine on the differential generation of activated Th cells belonging to different Th subsets is assessed. I do not mean to imply by this comment that such studies are insignificant or meaningless. But they leave open the question of whether the experimental conditions employed exist under physiological conditions, and if so, how do they arise? These considerations are why I feel the in vivo experiments in the mouse model of cutaneous leishmaniasis, described above and showing a pivotal role of IFN-γ [83] and IL-4 [82], respectively, in the generation of Th1 and Th2 cells, are so significant. 

There is a consensus that cytokines produced by most Th cells favor the further generation of Th cells belonging to the subset to which the Th cell producing the cytokine belonged. The cytokine achieves this result by either preferentially supporting the further generation of Th cells belong to its subset, or inhibiting the generation of Th cells belong to opposing subsets. Two examples illustrate this tendency. The IL-4 made by Th2 cells stimulates Th2 but not Th1 cells to divided [80]; the IFN-γ produced by Th1 cells inhibits the proliferation of Th2 but not of Th1 cells [81]. Given this pattern of activities, I have made a case elsewhere that the threshold mechanism initially operates to control whether Th1 and Th2 cells are predominantly generated, and that the Th cells belonging to the predominant Th subset become ever more predominant, due to the nature of the cytokines activities in self-promoting the further generation of Th cells belonging to the same Th subset that produced them, leading to greater coherence. I refer to this proposal as The Cytokine Implementation Hypothesis [72]. Three points are pertinent to this proposal. There is evidence that coherence increases as the immune response evolves [87], as anticipated in the hypothesis. We confirmed that the findings of others that the generation of Th2 cells is dependent on IL-4. We demonstrated that T cells themselves, rather than other cells, produce the critical IL-4 [86]. This finding is anticipated on the Cytokine Implementation but not on the Cytokine Milieu Hypothesis. Lastly, The Principle of Independence can be accounted for if antigen-specific B cells mediate the cooperation envisaged in the threshold mechanism, as different B cells will be mediating the cooperation for non-cross-reacting antigens [78].

## 15. Foundational Ideas and World Health

I must admit some frustration and puzzlement at how what I regard as questionable frameworks have remained so dominant in research-intensive fields. I have tried to elaborate the view here how important it is to treasure paradoxes. The primary question addressed here is surely pertinent to developing strategies of the prevention and treatment of disease in many areas of medicine. I am surprised that the importance of antigen dose in controlling the class of immunity generated is not more prominently recognized, despite many classical observations. In our laboratory, we have tested and found antigen dose or the number of slowly growing entities to be important whenever we have investigated its pertinence, namely in responses in mice to transplantable tumors, mycobacteria and the protozoan parasite, *L major*, as documented above. I have indicated above how knowledge of the importance of antigen dose is likely important in vaccination against pathogens and cancers preferentially susceptible to cell-mediated attacks. I have recently discussed ideas and supporting evidence that the level of antigen is also important in controlling the Th1/Th2 phenotype of chronic responses, and how such knowledge leads to proposals for personalized immunotherapy of HIV-1 infections [88] and personalized treatment of tuberculosis [89,90]. Allergies are another field of medicine in which the class of immunity generated is critical to whether the immune response is benign or leads to clinical pathology. However, in this case, in contrast to vaccination against and treatment of cancer and certain infectious diseases, the role of antigen dose in controlling the class of immunity is well recognized in the process of desensitization, whether natural or achieved through clinical intervention [91]. 

## 16. Conclusions

The primary question considered here is what determines whether antigen primarily stimulates the generation of Th1 or Th2 cells? Most detailed proposals are cast within the framework that different DAMP/PAMP signals, signals required to activate CD4 T cells, also determine the Th1/Th2 phenotype of the ensuing response. This framework is implausible, as it is inconsistent with diverse observations on the variables of immunization known to affect the Th1/Th2 phenotype of the ensuing response. An alternative, the Threshold Hypothesis, can, in contrast, account for these variables. It also provides a context for the design of strategies to prevent and treat clinical conditions where resistance against the causative entity is optimally realized by a sufficiently strong and predominant Th1 response. 

## Figures and Tables

**Figure 1 biology-12-01253-f001:**
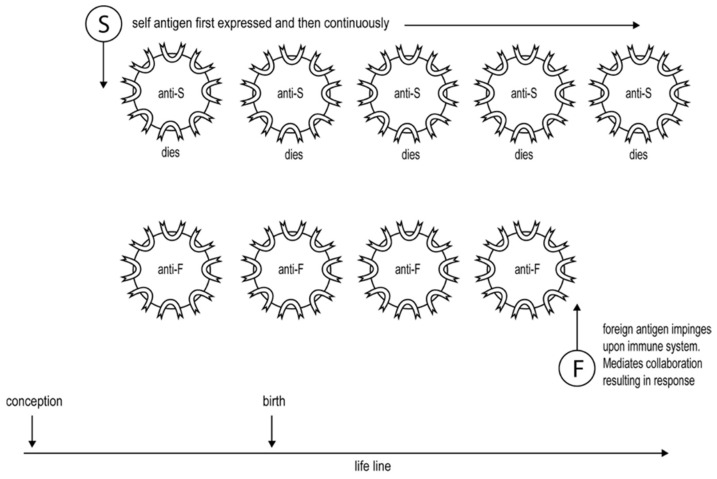
Illustration of how the requirement for the antigen-mediated cooperation of lymphocytes for their activation, but not for their inactivation, accounts for peripheral tolerance. Thus, lymphocytes specific for a self-antigen, S, see top row, are inactivated as generated one or a few at a time, whereas lymphocytes specific for a foreign antigen, F, accumulate in its absence, see bottom row. Once F impinges upon the immune system, it can mediate the lymphocyte cooperation to generate a response.

**Figure 2 biology-12-01253-f002:**
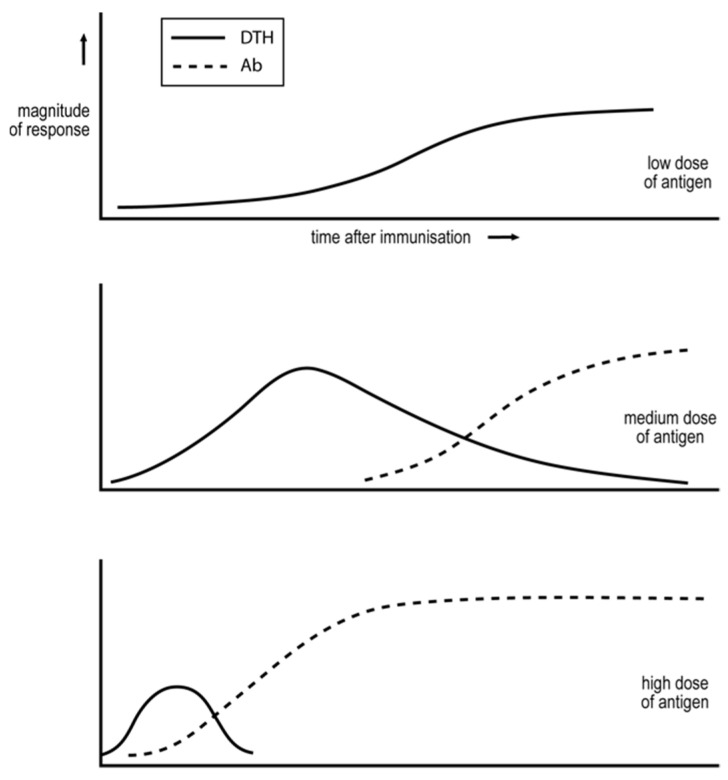
The dependence of the cell-mediated and IgG antibody response on antigen dose and time after immunization, as first revealed by the observations of Salvin [67]. The size of the cell-mediated response was assessed by measuring the strength of the DTH reaction.

## Data Availability

Not applicable.

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
