# Peer review of "What Determines the Class of Immunity an Antigen Induces? A Foundational Question Whose Rational Consideration Has Been Undermined by the Information Overload"

_biology, 2023, doi:10.3390/biology12091253_

Round 1

Reviewer 1 Report

The manuscript by Bretscher is a fundamental assay in basic immunology that aims to describe the current and past views and theories at the basis of the class immunity that develops in response to diverse antigens, namely Th1 versus Th2 responses. The manuscript is intriguing and well-written, builds on a large body of published research, including of the author’s own research. It first provides the context for this assay, and explains the broader significance of the selected topic. Then, there is a historical perspective of the experimental data and conclusions stemming from those, including the PAMP/DAMP versus the Quorom Model for Lym activation that reflects the personal view of the author based on given examples and explanations. Subsequently, there is a discussion on the importance of immune class regulation with selected examples, followed by a description of Salvin’s finding related to the variables that affect the phenotypic Th responses. Finally, the author dwells into his own views and models on the factors that control the Th1/Th2 responses with respect to current views and supporting/non-supporting evidences. Overall, the manuscript is comprehensive and supported by proper references, easy to read and thought provoking, the author also makes clear when his own views are less common.

Minor comments:

1.     Figure 1 is difficult to understand – consider revising or better explanation in figure legend (although after reading the relevant text in manuscript it is easier to follow, better explanation in the legend or an improved figure could better support reader understanding).

2.     In page 5 last sentence “in accord with the model”  - specify the referred model as several are discussed in the paragraph.

3.     Page 9, line 383  - correct Th1/Th1 to “Th1/Th2 phenotype”

Author Response

     I thank the reviewer for their careful reading and considered comments. I have increased the legend to Figure 1 in response to point 1 and made a correction in response to point 2. I have made the correction indicated by the reviewer in response to point 3. 

Reviewer 2 Report

I would like to thank the author for this contribution which is in my opinion very important. It stimulates a reflection on how to consider our personal research, and how to critically receive and interpret the profusion of increasingly sophisticated and specialized publications that we are experiencing nowadays.

The author has published several similar articles in the past years which are cited as references, each with a specific and complementary focus. I was just missing his recent contribution in Frontiers in Immunology (doi: 10.3389/fimmu.2022.960742), how does this new manuscript differs from this last article?

I have also a few questions/comments.

The quorum model/threshold hypothesis: the way the T lymphocytes cooperate in this model is not really explained: is a cell-cell contact necessary, which molecules & pathways are/could be involved, and how does this ultimately lead to Th1 or Th2 differentiation?

In the example of streptococcal infection (section 6, lines 248-253), I do not really understand why the DAMP/DAMP model would be more unlikely than the quorum model. Could the author explain this?

The Leishmania model discussed in sections 7 and 9 is very striking. The author discusses that genetic factors might be responsible for this. Does he essentially mean MHC polymorphism? Do mice strain with the same H-2 background have the same susceptibility to infection and similar “transition numbers”?

Another aspect which I was thinking of when reading the paper is the admitted plasticity of CD4 subsets. If the Th1 imprinting works, then it is unlikely that Th1 cells can become Th2 when challenging the animals with a high dose of the parasite: does the author think that CD4 Th plasticity is in general overestimated, or that specifically Th1/Th2 plasticity is unlikely?

Regarding the application of the author´s model to vaccination against cancer in patients. Current approaches rely on vaccination using several antigens (i.e. individual peptides from different antigens bearing HLA-class I and/or class II epitopes). Would the author also recommend to immunize first with very low amount of each HLA-class II antigen peptide to generate strong CD8 responses? How feasible it is to predict/test this amount (i.e to establish the “transition number”) for each single patient? And would this be differetn if using foreign e.g. mutated peptides vs overexpressed ones?

Minor

Regarding broad basics notions (e.g. immunology text books): the activation (priming) signal 2 is nowadays classically dissected in signal 2 (costimulation) and signal 3 (cytokines). I think it would be important to add this somewhere, especially for the young readership.

Line 383. Is Th1/Th2 meant (typo)?

Author Response

I am grateful to and thank reviewer 2 for their open-minded, empathetic reading of the manuscript. This is a rather rare experience! My comments to points raised below, and how I have amended the manuscript, are indicated in italics. I hope I might be able to discuss some issues with this reviewer once the paper is published.

I would like to thank the author for this contribution which is in my opinion very important. It stimulates a reflection on how to consider our personal research, and how to critically receive and interpret the profusion of increasingly sophisticated and specialized publications that we are experiencing nowadays.

The author has published several similar articles in the past years which are cited as references, each with a specific and complementary focus. I was just missing his recent contribution in Frontiers in Immunology (doi: 10.3389/fimmu.2022.960742), how does this new manuscript differs from this last article?

This article submitted to Biology is written for a more general audience than the Frontiers article, and so is considerably longer, and lays greater emphasis on how responses to simple and complex antigens can be similarly regulated, thus emphasizing the contrasting concepts of coherence of responses to different components of complex antigens and independence of responses to non-crossreacting antigens. I laid more emphasis on vaccination against entities best contained by cell-mediated attack, as I felt this would illustrate the potential importance of basic ideas.

I have also a few questions/comments.

The quorum model/threshold hypothesis: the way the T lymphocytes cooperate in this model is not really explained: is a cell-cell contact necessary, which molecules & pathways are/could be involved, and how does this ultimately lead to Th1 or Th2 differentiation?

The emphasis of the article is on considerations at the level of the system, in those areas where there is significant conflict at this level. As regards the Quorum model in the context of CD4 T cells, I do say several times that the cooperation between CD4 T cells is mediated by antigen-specific B cells. This mechanistic feature is central, as indicated, as to how independence of responses to non-cross-reacting antigens can be achieved. Moreover, as I explain, evidence supports this proposition. This minimally implies a three-cell interaction. We have addressed in other publications what CoS molecules are likely involved in mediating the CD4 T cell collaboration involved in determining Th1/Th2 phenotype. However, I have tried to focus the ideas and information in a particular way for this article, and feel going into this area would blunt the focus of the article. I have added the following to the manuscript in response to this comment: “One reviewer suggested it would be helpful to know how signal 2 is mediated in detail at the molecular level. I think signal 2 is surely different at the molecular level for different types of lymphocyte. In addition, it is not necessary to know such detail in testing The Two Signal Hypothesis, a proposal primarily at the level of the system. In addition, a mistaken or only partial identification of signal 2 in molecular terms may lead to observations that are erroneously interpreted as evidence against the hypothesis. It should be added for clarity that, in more recent times, the CoS signal is often referred to as signal 2 and the delivery of cytokines as signal 3.” 

In the example of streptococcal infection (section 6, lines 248-253), I do not really understand why the DAMP/DAMP model would be more unlikely than the quorum model. Could the author explain this?

I have added to the text in response to this comment the following: In particular, the Quorum Hypothesis explains why group A streptococci and heart tissue must crossreact at the level of CD4 T cells for CD4 T cell autoimmunity to be induced, whereas such crossreactivity is not anticipated to be necessary on the DAMP/PAMP Model. The incidence of autoimmunity would be expected to be much more prevalent on the latter model.

The Leishmania model discussed in sections 7 and 9 is very striking. The author discusses that genetic factors might be responsible for this. Does he essentially mean MHC polymorphism? Do mice strain with the same H-2 background have the same susceptibility to infection and similar “transition numbers”?

The reviewer is correct that MHC loci are involved, but resistance/susceptibility is a genetically complex trait governed by multiple loci. I really would like to avoid expanding on this topic in this manuscript as significant comments can be made that would, if included, dilute the focus I have attempted to realize. 

Another aspect which I was thinking of when reading the paper is the admitted plasticity of CD4 subsets. If the Th1 imprinting works, then it is unlikely that Th1 cells can become Th2 when challenging the animals with a high dose of the parasite: does the author think that CD4 Th plasticity is in general overestimated, or that specifically Th1/Th2 plasticity is unlikely?

A good point. This is again a complex topic bedeviled by considerations at the level of the system and at the cellular level. We have shown in a number of systems that the Th1 lock is mediated by antigen-specific CD8 T cells. The concept of plasticity is often primarily employed at the single cell level; ie can a Th1 cell be stimulated become a Th2 cell? I touched on plasticity in the more general sense, of whether a Th1/Th2 population can be modulated backwards into a predominant Th1 state. How does this occur? These are complex questions at the cellular and molecular level that should be addressed once it is accepted that they can occur at the level of a population of cells. I feel it sufficient at this stage to try to convince colleagues of the importance and existence of such in vivo modulations of populations of T cells.

Regarding the application of the author´s model to vaccination against cancer in patients. Current approaches rely on vaccination using several antigens (i.e. individual peptides from different antigens bearing HLA-class I and/or class II epitopes). Would the author also recommend to immunize first with very low amount of each HLA-class II antigen peptide to generate strong CD8 responses? How feasible it is to predict/test this amount (i.e to establish the “transition number”) for each single patient? And would this be differetn if using foreign e.g. mutated peptides vs overexpressed ones?  

Again, an interesting point. Personally, I think the ideal way of vaccination would be with vectors, such as BCG vectors, where it is possible to define conditions where the Th1/Th2 phenotype of the ensuing response can be assessed beforehand. We have shown the Th1/Th2 phenotype of the response against the vector-encoded antigen and the BCG antigens is coherent and is dependent on BCG vector dose. Vaccination with very low doses may give to exclusive Th1 responses to the vector in all individuals.

Minor

Regarding broad basics notions (e.g. immunology text books): the activation (priming) signal 2 is nowadays classically dissected in signal 2 (costimulation) and signal 3 (cytokines). I think it would be important to add this somewhere, especially for the young readership. Have addressed this in amended ms, see above.

Line 383. Is Th1/Th2 meant (typo)? Thanks, Corrected.

Reviewer 3 Report

Title: What determines the class of immunity an antigen induces? A foundational question whose rational consideration has been undermined by the information overload.

In this paper the authors study activated CD4 T helper cells are required to activate B cells to produce antibody and CD8 T cells to generate cytotoxic T lymphocytes. The authors describe CD4 T cell activation hypotheses and conclude that their alternative supports a rational approach to the prevention and treatment of several clinical conditions associated with infectious diseases and, more hypothetically, cancer.

This appears to be a review that hypothesizes mechanisms of activation of the immune system.

Figure legends should be more descriptive and longer.

The paper lacks conclusion

The paper goes into many parts, but fails to describe immune cells as "immune sentinels." An interesting article was recently published which I suggest you read, incorporate its meaning and bring back into discussion and references.

C. D’Ovidio. THE RESPONSE OF IMMUNE SENTINELS CAUSING INFLAMMATION IN GLIOMA AND GLIOBLASTOMA European Journal of Neurodegenerative Diseases 2023; 12(2): 46-50. (www.biolife-publisher.it)

In addition, the article talks about immunity and cancer. Authors should read the paper below and report the meaning in the text and list the article in the references.

E. Toniato.  IMMUNITY AND CANCER: IS THE VACCINATION READY FOR USE? European Journal of Neurodegenerative Diseases 2023; 12(1) :17-19. (www.biolife-publisher.it)

Without these changes the article cannot be published. MINOR REVISION.

It could be improved

Author Response

I thank the reviewer for their comments. My responses are given in italics.

In this paper the authors study activated CD4 T helper cells are required to activate B cells to produce antibody and CD8 T cells to generate cytotoxic T lymphocytes. The authors describe CD4 T cell activation hypotheses and conclude that their alternative supports a rational approach to the prevention and treatment of several clinical conditions associated with infectious diseases and, more hypothetically, cancer.

This appears to be a review that hypothesizes mechanisms of activation of the immune system.

I think I should say, with respect, that the above summary really does not reflect the substantial content of the submitted manuscript.

Figure legends should be more descriptive and longer. I have taken this comment into account.

The paper lacks conclusion.

I have included a Conclusion.

The paper goes into many parts, but fails to describe immune cells as "immune sentinels." An interesting article was recently published which I suggest you read, incorporate its meaning and bring back into discussion and references.

  1. D’Ovidio. THE RESPONSE OF IMMUNE SENTINELS CAUSING INFLAMMATION IN GLIOMA AND GLIOBLASTOMA European Journal of Neurodegenerative Diseases 2023; 12(2): 46-50. (www.biolife-publisher.it)

In addition, the article talks about immunity and cancer. Authors should read the paper below and report the meaning in the text and list the article in the references.

  1. Toniato. IMMUNITY AND CANCER: IS THE VACCINATION READY FOR USE? European Journal of Neurodegenerative Diseases 2023; 12(1) :17-19. (www.biolife-publisher.it)

 Without these changes the article cannot be published. MINOR REVISION.

    I have carefully read both of these recent and very short Opinion papers from EJND. I honestly could not find, on reading them, either novel ideas or observations bearing on the broad topic of the submitted manuscript. The idea of a role for sentinel cells is covered by the discussions on the DAMP/PAMP models. I acknowledge it is helpful if a reviewer brings to an author’s attention some pertinent literature of which the author may be unaware. I naturally read the papers that the reviewer suggested were significant. The reviewer stated: “Authors should read the paper below and report the meaning in the text and list the article in the references…..Without these changes the article cannot be published.” I must say I have never received such stark demands from a reviewer. I feel, if I took this advice, it would be detrimental to the paper, as I cannot discern novel observations or ideas in these papers. I do not consider that such a change, if implemented, would be a minor revision. I have therefore decided not to act on this advice.